# The COVID-19 Pandemic and Adolescents’ Psychological Distress: A Multinational Cross-Sectional Study

**DOI:** 10.3390/ijerph19148261

**Published:** 2022-07-06

**Authors:** Hang T. M. Nguyen, Hoang V. Nguyen, Btissame Zouini, Meftaha Senhaji, Kourosh Bador, Zsuzsa Szombathyne Meszaros, Dejan Stevanovic, Nóra Kerekes

**Affiliations:** 1Faculty of Psychology, University of Social Sciences and Humanities, Hanoi 10000, Vietnam; ntmhang@vnu.edu.vn; 2Department of Psychology, University of Minnesota, Minneapolis, MN 55455, USA; nguy2338@umn.edu; 3UAE/U24FS, FS, Abdelmalek Essaâdi University, Tetouan 90100, Morocco; btissamezouini@gmail.com (B.Z.); msenhaji@uae.ac.ma (M.S.); 4AGERA KBT, 41138 Gothenburg, Sweden; kourosh@meshe.se; 5Department of Psychiatry and Behavioral Sciences, SUNY Upstate Medical University, Syracuse, NY 13210, USA; meszaroz@upstate.edu; 6Department of Psychiatry, Clinic for Neurology and Psychiatry for Children and Youth, 11000 Belgrade, Serbia; stevanovic.dejan79@gmail.com; 7Department of Health Sciences, University West, 46132 Trollhattan, Sweden

**Keywords:** COVID-19 pandemic, brief symptom inventory, adolescents, gender, nationality

## Abstract

Background: The coronavirus disease 2019 (COVID-19) pandemic has continued for more than two years, and the impact of this pandemic on mental health has become one of the most important research topics in psychiatry and psychology. The aim of the present study was to assess psychological distress in adolescents across five countries (Sweden, Morocco, Serbia, Vietnam, and the United States of America) during the COVID-19 pandemic. Methods: Using nonparametric analyses we examined the impact of COVID-19 on distress, measured by the Brief Symptom Inventory, in a sample of 4670 adolescents. Results: Our results showed that the association between the COVID-19 impact and psychological distress in adolescents’ lives was positive and moderate in Morocco and Serbia, positive and weak in Vietnam and the United States of America, and negative and weak in Sweden. We also found that female adolescents reported higher distress levels than male adolescents. Conclusions: COVID-19 impacted adolescents and their psychological distress differently depending on their residence.

## 1. Introduction

The coronavirus disease 2019 (COVID-19) pandemic has continued for more than two years, and the impact of this pandemic on mental health has become one of the most important research topics in psychiatry, psychology, and public health. Specifically, cumulative results have led most researchers to believe that the COVID-19 pandemic and its associated restrictions, such as social distancing and lockdown, have created unprecedented hazards for mental health globally [1,2,3,4,5,6]. For instance, in a systematic review of 19 studies with a total sample of 93,569 adults from eight countries, (i.e., China, Denmark, Iran, Italy, Nepal, Spain, Turkey, and the United States of America [USA]), Xiong et al. [6] reported varying rates of symptoms of anxiety (6.33% to 50.9%), depression (14.6% to 48.3%), post-traumatic stress disorder (7% to 53.8%), and stress (8.1% to 81.9%) and high rates of psychological distress (34.43% to 38%). Providing further evidence for the negative impact of COVID-19 restrictions on the general population, other researchers have found increased rates of symptoms associated with psychiatric conditions, such as depression, anxiety, panic, and many others [7,8,9,10,11,12].

In addition to these recent studies on adult health, many researchers have investigated the effects of COVID-19 restrictions on children and adolescents, as they are more vulnerable to environmental stress and often have a more limited understanding of stressful situations and fewer strategies than adults to cope with sudden changes [13,14,15,16,17,18,19,20,21,22,23]. For instance, Meherali et al. [19] reviewed 18 studies with a total number of 20,150 children and adolescents from five countries, (i.e., Australia, Canada, China, Italy, and the USA) and concluded that COVID-19 restrictions had a great impact on the mental health of children and adolescents, causing anxiety, depression, disturbances in sleep and appetite, and impairment of social interactions. In addition, a review by Panda et al. [24] identified anxiety, depression, irritability, boredom, inattention, and fear of the COVID-19 pandemic as predominant psychological problems among children and adolescents.

In the following section, we provide an overview of the possible relationships between COVID-19 and adolescents’ psychological distress levels, focusing on nine domains: somatization, obsession–compulsion, interpersonal sensitivity, depression, anxiety, hostility, phobic anxiety, paranoid ideation, and psychoticism.

### 1.1. Somatization

Somatization was among the most frequent psychological problems observed during the COVID-19 pandemic. Specifically, researchers have indicated that adolescents reported more somatization symptoms during the pandemic than before the pandemic [8]. Common somatic complaints included myalgia and dizziness [5], headaches, sleeping problems [25], and stomachaches [26]. Furthermore, empirical evidence suggests that risk factors related to somatization symptoms include female gender [8,11], small living space, and family conflicts [26].

### 1.2. Obsessive–Compulsive Symptoms

Multiple studies have described worsening obsessive–compulsive disorder (OCD) symptoms during the COVID-19 pandemic [27]. Preventive measures associated with COVID-19, such as frequent hand washing, disinfection, and mask mandates, can have serious implications for those who are preoccupied with cleanliness and hygiene or suffer from OCD symptoms [28]. For instance, in their survey of children and adolescents in Canada, Cost et al. [29] reported that OCD rates varied across different age groups and preexisting psychiatric diagnostic groups, ranging from 13% to 30%. In another study that included 61 adolescents with OCD as their primary diagnosis, Tanir et al. [30] found a significant increase in symptom severity, (i.e., the most common symptoms were contamination obsessions and cleaning/washing compulsions) during the pandemic. The fear of COVID-19 shows a strong direct effect on OCD in adolescents (36%), and indirect effects through emotional reactivity, depression–anxiety, and experiential avoidance [31]. A recent study by Nissen et al. [32] reported that diagnosed group of children and adolescents with OCD symptoms described that thoughts about COVID-19 became an integral part of their OCD. Several other studies have also reported the impact of the COVID-19 pandemic on OCD symptoms [33,34].

Several factors correlated positively with OCD symptom severity during the COVID-19 pandemic, including talking and online searching about COVID-19, daily preoccupation with COVID-19, duration of OCD diagnosis, diagnosis of COVID-19 in a friend or relative, and family history of attention deficit hyperactivity disorder [30,32].

### 1.3. Interpersonal Sensitivity

Although there are only a limited number of studies published on interpersonal sensitivity, few authors [35] have reported an upward trend in this problem. Specifically, in a recent study of Chinese adolescents, Qin et al. [35] found that sensitivity tendency increased from 19.8% before to 46% during the COVID-19 pandemic. Furthermore, this study also found that risk factors included female gender, no siblings, and exposure to COVID-19 due to family members testing positive.

### 1.4. Depression

According to recent studies, the prevalence of depression in adolescents during the COVID-19 pandemic is higher than it was before the pandemic [8,20,22,23,36,37]. For instance, Serra et al. [37] reported that approximately 24% of children and adolescents in Italy reported typical depressive symptoms, such as depressed mood, physical symptoms, and negative self-esteem. The rate of depression among Chinese adolescents was found to be between 22.6% [22] and 43.7% [23], including the common symptoms of little interest or pleasure in doing things, feeling tired or having little energy, poor appetite, and overeating. Other studies reported an even higher prevalence of depressive symptoms in adolescents, for example, 51% in a Canadian sample [15] and 71.5% in a Chinese sample [38].

Regarding the risk factors for depression, many studies have found that female gender [23,39], poor academic performance [38], and older age were associated with a higher probability of depression. Notably, Hawes et al. [8] found that among female participants, there was a nearly three-fold increase in rates of clinically significant depression from the pre-COVID-19 period to the COVID-19 period. With regard to protective factors against depression, some authors suggest that frequent physical exercise [38,40] and frequent communication with parents [41] were among the key factors.

### 1.5. Anxiety Symptoms

Anxiety in adolescents during COVID-19 has been investigated by many researchers. Results from several studies with international samples and different anxiety measurements suggest that the frequency and severity of anxiety among adolescents have increased compared to before the pandemic [8,9,35,42,43]. For instance, in a survey conducted on 1480 children and adolescents in Italy, Spain, and Portugal by Francisco et al. [44], approximately one-third of participants felt restless, nervous, worried, uneasy, and anxious; more than half felt bored (52.2%), and one-third felt lonely. In another study that included 8079 Chinese adolescents, Zhou et al. [23] indicated that 37.4% reported mild to severe anxiety. The most common symptoms were feeling nervous, anxious, or on edge, worrying too much about different things, and becoming easily annoyed or irritable [23]. It is noteworthy that other authors [38] reported an even higher proportion of adolescents with anxiety (54.5%) in China. Furthermore, such relatively high anxiety rates have also been observed in other countries, such as Brazil (19.4%) [45] and Canada (39%) [15]. Interestingly, in a recent study that investigated anxiety in Moroccan high schoolers, Mzadi et al. [20] found that these students had lower anxiety compared with the pre-pandemic period. When examining the relationship between anxiety and emotional awareness, Smirni et al. [43] found that anxiety did not seem to be associated with emotional awareness and management, which supports the hypothesis that the COVID-19 pandemic and its preventive measures increased the anxiety rate.

Regarding the risk factors for anxiety, several studies have suggested that female gender [11,23,40,42,43], parents with high anxiety [42], exposure to excessive information, closure of schools, and home quarantine [8,46], small family unit, (i.e., stem family and no siblings), and risk of exposure to COVID-19 due to family members testing positive [35] were associated with higher rates of anxiety. A few authors also suggested that protective factors against anxiety could include physical exercise [35,40] and having a more extroverted personality [35].

### 1.6. Hostility

Two studies have found that the COVID-19 pandemic caused increased hostility levels among young people [47,48]. In a study of 549 high schoolers, Karaman et al. [49] found a correlation between the impact of COVID-19 and adolescents’ hostility (r = 0.62). It was notable that, on average, female adolescents had higher hostility scores than their male counterparts.

### 1.7. Phobic Anxiety Symptoms

As with other distress problems, phobic anxiety also increased in adolescents during the pandemic compared to the pre-pandemic period [35]. Risk factors included female gender and living in a nuclear family [35], whereas daily exercise was reported as a protective factor [35].

### 1.8. Psychoticism and Paranoid Ideation

A recent study by Mzadi et al. [20] investigated psychoticism in Moroccan adolescents. They found that there was only a negligible difference between students’ psychoticism measurements before and during COVID-19. Among students who participated during the COVID-19 pandemic, this study found that female adolescents scored significantly higher on the psychoticism measure. Another study by [50], found that adolescents who were more stressed displayed more psychotic-like experiences during the pandemic. Furthermore, good family functioning, such as adaptation, partnership, growth, affection, and resolution, were protective factors against psychotic-like experiences.

Differences in paranoid ideation among adolescents before and during COVID-19 in Morocco were also investigated by Mzadi et al. [20]. Specifically, these authors found a significant difference in paranoid ideation between the two groups of adolescents, suggesting that the COVID-19 pandemic and government measures could have negatively impacted Moroccan adolescents. Additionally, Bauer et al. [51] published several case studies in which American adolescents with COVID-19 developed delusions and paranoid thoughts.

In summary, based on prior studies, the COVID-19 pandemic had a strong impact on adolescents’ distress, resulting in increased somatization, obsession–compulsion, depression, anxiety, and paranoid ideations, and worsened interpersonal sensitivity. Risk factors for adolescent mental health problems were primarily environmental, such as isolation at home, living in a nuclear family, low socioeconomic status, risk of family members’ exposure to COVID-19, overcrowded living space, and family conflict. Furthermore, personal risk factors for mental health problems included female gender, talking and online searching about COVID-19, home quarantine, and fear that family members could be infected. Protective factors against distress problems included physical exercise, frequent communication with parents, extroversion, positive refocusing, and cognitive reappraisal.

Although the above studies have provided initial insight into the impact of the COVID-19 pandemic on adolescent psychological distress, very few were cross-cultural studies and did not recruit participants from different countries. This limits our ability to develop a global perspective on the psychological distress of adolescents. For instance, what is the global impact of the COVID-19 pandemic on adolescent mental health? How is the impact similar or different in adolescents from different cultures? Can the global impact of COVID-19 restrictions on adolescents be determined? We hypothesized that the COVID-19 pandemic had an overall, global impact on adolescents’ psychological distress, but that the impact on various psychological symptoms may vary across different countries.

To answer the above questions and to test the hypothesis, the present study focused on evaluating levels of psychological distress in adolescents across five countries during the COVID-19 pandemic.

## 2. Materials and Methods

The research presented here is a cross-sectional study conducted in five countries on four continents: Sweden, Morocco, Serbia, Vietnam, and the USA. In each country, a coauthor obtained a local ethical review board’s approval before administering an online survey (available in five languages) to high school students, who were recruited either by their schools or via social media. The survey, available from September 2020 to February 2021, was voluntary and anonymous. A detailed description of this comprehensive study can be found in [52].

### 2.1. Measures

The Brief Symptoms Inventory (BSI) [53] is a multidimensional tool for mental health status assessment. The BSI consists of 53 items that assess somatic, emotional, and behavioral symptoms/functions across nine domains: somatization (7 items), obsession–compulsion (6 items), interpersonal sensitivity (4 items), depression (6 items), anxiety (6 items), hostility (5 items), phobic anxiety (5 items), paranoid ideation (5 items), and psychoticism (5 items). In addition to these domains, the BSI includes four independent items that provide important clinical information about the presence of conscious feelings of guilt, poor appetite, trouble falling asleep, and thoughts of death or dying. Participants in the study were asked to rank the intensity of their distress for each of the BSI items during the past seven days on a 5-point Likert scale from 0 (not at all) to 4 (extremely). Finally, item scores were summed to construct corresponding domain scores and averaged to form the Global Severity Index (GSI), which measured participants’ general psychological distress levels.

The COVID-19 impact: To measure the overall impact of the COVID-19 pandemic on their lives, all participants were asked to rate the following item “Please think about and assess how much the COVID-19 outbreak has personally affected you, your daily routines, work, and your family life?” on an 11-point visual analog scale from 0 (has not affected me at all) to 10 (has affected me immensely).

### 2.2. Participants

In total, 5114 adolescents (aged 15–19 years) responded to the BSI from five countries (Sweden, Morocco, Serbia, Vietnam, and the USA). Approximately 76% answered all items, 15% skipped 1 to 5 items, 2% neglected to answer 6 to 50 items, and 7% omitted more than 50 items (out of a total of 56 items). Based on these percentages, we analyzed data from 4670 (91%) participants who had at most five missing values, which were imputed using the predictive mean matching method [54]. Table 1 shows the general demographic information of the study sample.

### 2.3. Data Analyses

As a first step, we investigated the measurement invariance of the BSI subscales across the countries using multi-group confirmatory factor analysis (MG-CFA) [55] to document its cross-cultural validity, since no such data was available. For each subscale of the BSI, we fitted a confirmatory factor analysis (CFA) overall model using relevant items before fitting the configural, metric, and strong invariance models. Each subscale had to, at a minimum, pass metric invariance to be considered in further analyses. Model parameters were estimated using the maximum likelihood method. The absolute model fit to the data was evaluated using the Tucker–Lewis index (TLI), comparative fit index (CFI), standardized root mean square residual (SRMR), and root mean square error of approximation (RMSEA) with the following cut-off points: TLI and CFI ≥ 0.90, SRMR and RMSEA ≤ 0.08 as adequate and TLI and CFI ≥ 0.95, SRMR and RMSEA ≤ 0.06 as good fit [56,57]. A change in CFI and RMSEA values was used to judge the difference in the fit between nested models and values ≤ 0.01 for at least one of them were considered to indicate the invariance of the models [58].

To test for differences in the levels of psychological distress domains across the countries, we performed nonparametric multiple comparisons [59] that controlled for the family-wise error rate. Furthermore, to marginalize the influence of sample sizes on our results, (i.e., the five countries had various sample sizes), for each analysis, the Kruskal–Wallis test was conducted using the pseudo-rank rather than the commonly used rank [60].

We only examined male and female differences, as less than 2% of participants identified themselves as gender non-binary. To test for differences in levels of psychological distress domains across genders (male/female) we used the nonparametric Wilcoxon–Mann–Whitney test [60]. Next, we conducted Spearman correlation tests among the BSI subscales and COVID-19 impact scores. Finally, to examine the influence of the COVID-19 pandemic on mental health, for each country, we performed a regression analysis using the GSI score as a dependent variable and COVID-19 and gender as independent variables. All data analyses were carried out using the R program [61].

## 3. Results

### 3.1. Measurement Invariance of the Brief Symptoms Inventory Domains

Table 2 summarizes the goodness of fit for the overall, configural, and metric models tested. With the exception of the hostility domain’s CFA models, all models showed a good fit to the data. Relatively high values of RMSEA in some models, (i.e., obsession–compulsion, interpersonal sensitivity, depression, anxiety) were expected, as these models have a small degree of freedom [62]. Thus, we concluded that all domains, with the exception of the hostility domain, had the same factor structure across countries. In the remaining analyses, the hostility domain was excluded.

### 3.2. Psychological Distress across the Five Countries

Table 3 summarizes the differences in scores between the five countries in regard to the BSI domains, the GSI, and the COVID-19 restriction impact. Note that the GSI score was computed excluding the hostility domain’s items. In general, American and Moroccan adolescents reported significantly (*p* < 0.001) higher psychological distress levels than those from Serbia, Sweden, and Vietnam. Furthermore, our results show that American and Moroccan adolescents reported being significantly more impacted by the COVID pandemic (*p* < 0.001) than their counterparts in Serbia, Sweden, and Vietnam.

### 3.3. Psychological Distress and Gender Differences

The Wilcoxon–Mann–Whitney test results (see Table 4) show that the relative effect p(1,2) of the two independent samples 1 (male) and 2 (female) is significantly (*p* < 0.001) larger than 0.5 across all BSI domains and GSI scores, indicating that female adolescents reported significantly higher levels of psychological distress than male adolescents. Furthermore, our results showed that female adolescents reported being more affected by COVID-19 restrictions than their male counterparts (p(1,2) = 0.54, *p* < 0.001).

### 3.4. Psychological Distress and COVID-19 Impact

Table 5 shows correlations between the impact of COVID-19 score and BSI (with the exclusion of the hostility domain) and GSI scores by country and gender. Our results showed that among Swedish adolescents, BSI and GSI scores were negatively correlated with the impact of the COVID-19 score. Nonetheless, these correlations were relatively weak, with the highest correlation between COVID-19 impact and depression, (i.e., *r* = −0.28, *p* < 0.001 for males; *r* = −0.25, *p* < 0.001 for females). With regard to Moroccan adolescents, our results showed that BSI and GSI scores were positively correlated to the COVID-19 impact score. We also found that in the Moroccan sample, the highest correlation was between COVID-19 impact and interpersonal sensitivity for males (*r* = 0.47, *p* < 0.01) and between COVID-19 and somatization for females (*r* = 0.39, *p* < 0.001). In Serbian adolescents, our results showed that the COVID-19 impact had the strongest correlations with depression in females (*r* = 0.35, *p* < 0.001) and with phobic anxiety in males (*r* = 0.22, *p* < 0.001). Our results also showed that in Vietnamese adolescents, the correlation between COVID-19 impact and BSI domains ranged between 0.08 and 0.15 (*p* < 0.05) for both genders, which indicates a relatively weak relationship between COVID-19 and psychological distress (as conceptualized through the eight BSI domains). Finally, for American adolescents, our results show that COVID-19 impact was significantly correlated with depression (*r* = 0.24, *p* < 0.001) and psychoticism (*r* = 0.21, *p* < 0.01) in females and with paranoid ideation (*r* = 0.25, *p* < 0.05) in males.

### 3.5. COVID-19 Impact as a Predictor of Psychological Distress

We explored the effect of the COVID-19 impact and gender on the overall psychological distress, as measured by the GSI in five countries (Table 6).

Gender was a significant predictor of the overall psychological distress for Moroccan and Vietnamese adolescents only. Additionally, although the COVID-19 impact score predictor was significant in all regression models, the multiple *R*^2^, (i.e., the proportion of the variance in the GSI score that was accounted for by the variance in COVID-19 impact) of Sweden, Vietnam, and USA models were relatively small (approximately 5%). In other words, the COVID-19 variable was a better predictor of psychological distress for Moroccan, (i.e., *R*^2^ = 0.17) and Serbian adolescents, (i.e., *R*^2^ = 0.13). Finally, our results showed that a higher COVID-19 score predicted a higher GSI score for adolescents from Morocco, Serbia, Vietnam, and the USA (*β* = 0.12, =0.10, =0.03, =0.07, *p* < 0.001) and a lower GSI score for Swedish adolescents (*β* = −0.08, *p* < 0.001).

## 4. Discussion

The present study investigated adolescents’ psychological distress during the COVID-19 pandemic in five countries on four continents (Sweden, Morocco, Serbia, Vietnam, and the USA). The analyzed data were self-reported by 4670 adolescents. Nine domains of the BSI were explored (somatization, obsession–compulsion, interpersonal sensitivity, depression, anxiety, hostility, phobic anxiety, paranoid ideation, and psychoticism). We found significant cross-cultural and gender differences in psychological distress, with the impact of COVID-19 being different across the five countries. These results supported our hypothesis that although the COVID-19 pandemic had a global impact on adolescents’ psychological distress, the impact on various psychological symptoms may vary across different countries.

First, our results provided evidence for the cross-cultural validity of the eight BSI domains, but not the hostility domain. This suggests that BSI is a valid cross-cultural measurement tool for psychological distress, except for hostility/aggressive behaviors. Our results suggest that hostility is perceived differently in different cultures, with different social norms and approaches being used to regulate opposition. It has been suggested previously that aggressive and antisocial behaviors cannot/should not be assessed cross-culturally [63]. For instance, two out of five items from the hostility domain asked adolescents about the frequency of outbursts and arguments. Although such overt behaviors may be acceptable in several countries, Vietnamese adolescents, for example, are expected to avoid such behaviors to maintain social harmony. Thus, a review of the BSI’s hostility domain may benefit from considering a multidimensional construct view that includes verbal aggression, physical aggression, hostile affect, covert aggression, and bullying [63,64].

Second, our results revealed cultural differences in adolescents’ psychological distress during the COVID-19 pandemic. According to our data, American and Moroccan adolescents were most impacted by COVID-19, whereas Swedish and Vietnamese adolescents were the least affected. Serbian adolescents scored between these two extremes. These differences could be partially explained by the different pandemic situations in these countries at the time of data collection. For instance, during the data collection period, Sweden and Vietnam had lighter restrictions than other countries; Sweden had a unique approach to COVID-19, and Vietnam successfully handled their first two COVID-19 waves. Furthermore, high COVID-19 infection rates and prolonged lockdowns in Morocco, Serbia, and the USA at the time of data collection undoubtedly had a major impact on the responses of adolescents from these countries.

With regard to psychological distress, American and Moroccan adolescents reported the highest distress levels, whereas Serbian and Vietnamese adolescents scored the lowest. Swedish adolescents ranked between these two extremes. These results reflect a partial mismatch between the COVID-19 impact and the distress rankings between these countries. Serbia and Sweden switched places in the two measures, which suggests that Swedish adolescents may have a higher “basic” level (not situational, caused by the restrictions of the COVID-19 pandemic) of psychological distress than adolescents from Vietnam and Serbia. It is difficult to disentangle if the reportedly high impact of the COVID-19 pandemic on adolescents’ lives in Morocco and the USA is the main reason that these two countries’ adolescents also reported the highest level of psychological distress or that a higher “basic” level of psychological distress (mirroring cultural and national differences) may explain the two countries’ rankings. Other studies have focused on and investigated mediator variables, such as coping strategies, social isolation, social support from family and friends, access to the internet, and the use of social media, for their direct effect on adolescents’ psychological distress during the pandemic [65,66,67,68]. Li et al. [69] showed that problem-based coping and online learning satisfaction fostered adolescents’ adjustment both directly and indirectly, serving as a buffer against the negative impact of stressors on adjustment. Their findings support our result that Swedish adolescents—who live in a society with well-functioning online education, stable internet connections, and school-provided computers—reported experiencing the lowest impact from the COVID-19 pandemic.

Vietnamese adolescents also reported a low impact of the pandemic on their everyday lives in our study, while also scoring relatively low on psychological distress. This may be explained mostly by cultural impacts on adolescents’ lives. Specifically, in addition to Buddhist practices that have been common among the Vietnamese people for many centuries [70], ancestor worship is a common cultural and folk-religion practice [71]. Most Vietnamese people believe that their dead relatives are still present in their lives and that they possess the spiritual ability to help them. Thus, in difficult situations, Vietnamese people often offer incense to their ancestors’ altars and pray to them with the hope and belief that their dead ancestors will bless, guide, and protect them from misfortunes. For the Vietnamese people, ancestor worship can be valuable as a ready source of comfort in times of need [72]. It is worth noting that praying and/or meditation have been shown to reduce the probability of mental health problems in adolescents [73]. That religious beliefs are not necessarily protection against heightened psychological distress levels was indirectly proven in our study, in which Moroccan adolescents, also from a culture with a strong religious influence, still reported one of the strongest impacts of COVID-19, as well as the highest level of psychological distress. To demonstrate the complexity of the factors influencing adolescents’ mental health, we should mention that family problems and parents’ stress were associated with worsened mental health in adolescents during the COVID-19 pandemic [73,74,75] and that one of the countries where significantly increased parental stress levels were measured from pre-COVID to the first and second waves COVID periods was the USA [76].

Third, our results also showed that female adolescents (in the multinational sample) were more likely to report stronger COVID-19 impacts and psychological distress than male adolescents. These findings are consistent with previous studies, where female adolescents reported higher levels of distress [11,23,35,39,40,42,43,49]. This can be partially explained by known gender differences in interpersonal communication and social responsibility. For instance, early research on gender differences in communication showed that, in general, women are more dependent on social emotions through their contact with other people than men [77,78,79,80]. Furthermore, later works on gender differences [81,82,83,84] found that females use language to deepen their social connections and create new relationships, whereas men communicate to increase social dominance and achieve tangible outcomes. In line with these results, several researchers [85,86] have argued that men communicate to establish common activities, whereas women value the process of communication itself; that is, women simply want to share their stories and thus tend to communicate face-to-face [87]. Considering these results, we suggest that the COVID-19 pandemic, which led to social distancing (and lockdown in many countries), impeded face-to-face communication and caused more psychological distress to females than males.

Another explanation for gender differences in levels of psychological distress can also stem from gender inequality in housework (work without compensation), the primary responsibility for which typically falls on women and adolescent girls [87,88]. For instance, many researchers have documented that in many countries, male adolescents do not have to do as much housework as female adolescents [89,90,91,92,93]. It could be the case that during the COVID-19 lockdown in many countries, staying at home also increased housework responsibilities for female adolescents and thus their psychological distress.

Lastly, we found that the correlation between COVID-19 impact and psychological distress was moderate in Morocco and Serbia, but weak in Sweden, Vietnam, and the USA. Interestingly, COVID-19 impact correlated positively with distress in all countries, except in Sweden, where COVID-19 correlated negatively with all distress indicators among Swedish adolescents. However, these correlation coefficients are low, (i.e., their absolute values are less than 0.30) and suggest no clinical implications. A recent longitudinal study [14] found no differences between Swedish adolescents who were and were not exposed to COVID-19 in terms of mental health, relationships with parents and peers, and health behaviors. These results suggest that Swedish adolescents had high levels of resilience and effective coping strategies during the pandemic. This may be related to culture, a healthy lifestyle, higher socioeconomic status, better access to the internet, better social and welfare organizations, and less isolation. Thus, a major stressor, such as the COVID-19 pandemic, could have stimulated positive coping skills in Swedish adolescents. However, COVID-19 may have also impacted Swedish adolescents in other ways that were not captured by the BSI.

### Limitations

Like all cross-sectional studies, the present study has several limitations. For instance, our samples were not representative, as our participants were drawn from selected regions with a couple of large cities within the participating countries, except in Sweden (where all counties were represented), which could have led to a sampling bias of adolescents from a higher socioeconomic status. The number of participants per country was also variable. In addition, our dataset included only a handful of non-binary adolescents, which prevented us from making any meaningful comparisons between them and those belonging to other gender groups. Note that, although there were more female than male adolescents in our samples, our statistical analyses ensured that the uneven female-to-male ratio did not affect our results. Third, when evaluating the COVID-19 impact, we used only one simple question on personal feelings about the impact, operationalized in a scale, rather than a validated instrument, and thus might not have reliably captured the impact of COVID-19 restriction on adolescents’ lives. Finally, although the present research investigated the impact of the COVID-19 pandemic on adolescents, we did not examine other risk and protective factors aside from gender. Further studies are needed to examine the effects of race, ethnicity, gender identity, sexual orientation, and socioeconomic status on mental health outcomes in adolescents.

## 5. Conclusions

The present study examined the impact of COVID-19 on psychological distress among 4670 adolescents from five countries across four continents (Sweden, Morocco, Serbia, Vietnam, and the USA). We found that American and Moroccan adolescents were impacted the most by the COVID-19 pandemic and reported the highest levels of distress, whereas Vietnamese adolescents were the least affected by COVID-19 and had the lowest levels of distress. Furthermore, we also found that the COVID-19 impact correlated positively with psychological distress in all countries except Sweden. Finally, we found that female adolescents had higher COVID-19 impact scores and reported more psychological distress than male adolescents.

## Figures and Tables

**Table 1 ijerph-19-08261-t001:** General demographic information of the study sample.

	Gender			Age in Years
Sample *(n)*	Male, *n (%)*	Female, *n (%)*	Non-Binary, *n (%)*	*M (SD)*
Total (4670)	1685 (36.1)	2927 (62.7)	58 (1.2)	16.70 (1.02)
Sweden (1525)	579 (38.0)	932 (61.1)	14 (0.9)	17.16 (0.88)
Morocco (484)	172 (35.5)	308 (63.6)	4 (0.9)	16.83 (1.21)
Serbia (1065)	351 (33.0)	707 (66.3)	7 (0.7)	16.31 (1.05)
Vietnam (1305)	521 (39.9)	768 (58.9)	16 (1.2)	16.45 (0.86)
USA (291)	62 (21.3)	212 (72.9)	17 (5.8)	16.60 (0.99)

Notes. *n* = sample size, *M* = sample mean, and *SD* = sample standard deviation.

**Table 2 ijerph-19-08261-t002:** Measurement invariance of the BSI domains across five countries.

Subscale	Model	α*/*ω	χ^2^ (df)	CFI	TLI	RMSEA	SRMR	ΔCFI/ΔRMSEA
SO	Overall	0.86/0.86	137.14 (14)	0.98	0.98	0.06	0.01	-
	Configural		244.70 (70)	0.97	0.96	0.07	0.03	0.01/−0.01
	Metric		348.15 (94)	0.96	0.96	0.07	0.05	0.01/0
OC	Overall	0.85/0.85	304.15 (9)	0.96	0.94	0.10	0.03	-
	Configural		372.40 (45)	0.95	0.92	0.10	0.04	0.01/0
	Metric		484.12 (65)	0.95	0.94	0.09	0.05	0/0.01
IS	Overall	0.83/0.83	86.50 (2)	0.98	0.95	0.12	0.02	-
	Configural		94.67 (10)	0.98	0.95	0.12	0.02	0/0
	Metric		250.65 (22)	0.96	0.95	0.12	0.06	0.02/0
DE	Overall	0.89/0.90	440.40 (9)	0.96	0.93	0.12	0.03	-
	Configural		602.80 (45)	0.94	0.91	0.14	0.04	0.02/−0.02
	Metric		767.52 65)	0.94	0.93	0.12	0.06	0/0.02
AN	Overall	0.88/0.88	283.10 (9)	0.97	0.95	0.10	0.03	-
	Configural		417.88 (45)	0.95	0.93	0.12	0.03	0.02/−0.02
	Metric		632.26 (65)	0.94	0.93	0.12	0.07	0.01/0
HO	Overall	0.82/0.83	489.87 (5)	0.90	0.80	0.19	0.05	-
	Configural		619.76 (25)	0.88	0.76	0.21	0.06	0.02/−0.02
	Metric		776.32 (41)	0.86	0.83	0.18	0.08	0.02/0.03
PA	Overall	0.79/0.80	41.55 (5)	0.99	0.98	0.05	0.02	-
	Configural		104.03 (25)	0.98	0.95	0.08	0.02	0.01/−0.03
	Metric		181.92 (41)	0.96	0.95	0.08	0.05	0.02/0
PI	Overall	0.81/0.82	69.07 (5)	0.99	0.97	0.06	0.02	-
	Configural		102.84 (25)	0.98	0.96	0.07	0.02	0.01/−0.01
	Metric		143.54 (41)	0.97	0.97	0.06	0.03	0.01/0.01
PS	Overall	0.78/0.80	129.98 (5)	0.97	0.94	0.09	0.03	-
	Configural		141.84 (25)	0.97	0.94	0.09	0.03	0/0
	Metric		228.97 (41)	0.96	0.95	0.08	0.05	0.01/−0.01

Notes. BSI = Brief Symptom Inventory, SO = somatization, OC = obsession–compulsion, IS = interpersonal sensitivity, DE = depression, AN = anxiety, HO = hostility, PA = phobic anxiety, PI = paranoid ideation, and PS = psychoticism. The overall model denotes the overall factor model, the configural model tests the configural invariance, the metric model tests the weak invariance, α denotes the Cronbach’s alpha for the total sample, and ω denotes the McDonald’s omega for the total sample size. CFI = comparative fit index, TLI = Tucker–Lewis index, RMSEA = root mean square error of approximation, SRMR = standardized root mean square residual, and ΔCFI/ΔRMSEA = negative difference between a model index and that of the model immediately above.

**Table 3 ijerph-19-08261-t003:** Psychological distress differences across five countries.

	Sweden (1)		Morocco (2)		Serbia (3)		Vietnam (4)		USA (5)		Difference	
	*M*	*SD*	*M*	*SD*	*M*	*SD*	*M*	*SD*	*M*	*SD*	*p*-Value	Post-Hoc
SO	1.00	0.82	1.26	0.88	0.74	0.80	0.59	0.68	1.22	0.97	<0.001	4 < 3 < 1 < 2; 4 < 3 < 1 < 5
OC	1.71	0.94	1.77	0.90	1.10	0.96	1.16	0.87	2.24	0.97	<0.001	4 < 1 < 5; 3 < 1 < 5; 4 < 2 < 5; 3 < 2 < 5
IS	1.46	1.08	1.55	0.96	1.07	1.10	1.25	1.07	2.13	1.11	<0.001	3 < 4 < 1 < 5; 3 < 4 < 2 < 5
DE	1.74	1.06	1.52	0.98	1.03	1.02	1.06	0.96	2.19	1.04	<0.001	3 < 2 < 1 < 5; 4 < 2 < 1 < 5
AN	1.38	0.95	1.40	0.97	0.93	0.95	0.74	0.77	1.60	1.07	<0.001	4 < 3 < 1; 4 < 3 < 2; 4 < 3 < 5
PA	0.94	0.91	1.24	0.91	0.69	0.81	0.69	0.71	1.40	1.05	<0.001	4 < 1 < 5; 3 < 1 < 5; 4 < 1 < 2; 3 < 1 < 2
PI	1.26	0.98	1.80	0.96	1.15	0.94	0.73	0.76	1.62	0.99	<0.001	4 < 3 < 1 < 2; 4 < 3 < 1 < 5
PS	1.03	0.87	1.30	0.94	0.84	0.92	0.82	0.84	1.66	0.94	<0.001	3 < 1 < 2 < 5; 4 < 1 < 2 < 5
GSI	1.28	0.76	1.43	0.80	0.94	0.80	0.83	0.71	1.71	0.81	<0.001	4 < 3 < 1 < 2 < 5
COV	3.48	2.42	5.32	2.97	4.83	3.06	4.37	2.89	7.32	2.36	<0.001	1 < 4 < 3 < 2 < 5

Notes. SO = somatization, OC = obsession–compulsion, IS = interpersonal sensitivity, DE = depression, AN = anxiety, HO = hostility, PA = phobic anxiety, PI = paranoid ideation, and PS = psychoticism, GSI = Global Severity Index, COV = COVID.

**Table 4 ijerph-19-08261-t004:** Psychological distress differences by gender.

	Effect	Estimator	Std Error	Statistic	*p*-Value
SO	p(1,2)	0.648	0.009	16.810	<0.001
OC	p(1,2)	0.622	0.009	13.835	<0.001
IS	p(1,2)	0.645	0.009	16.512	<0.001
DE	p(1,2)	0.606	0.009	12.079	<0.001
AN	p(1,2)	0.661	0.009	18.266	<0.001
PA	p(1,2)	0.654	0.009	17.565	<0.001
PI	p(1,2)	0.605	0.009	11.954	<0.001
PS	p(1,2)	0.585	0.009	9.632	<0.001
GSI	p(1,2)	0.639	0.009	15.806	<0.001
COV	p(1,2)	0.538	0.009	4.303	<0.001

Notes. SO = somatization, OC = obsession–compulsion, IS = interpersonal sensitivity, DE = depression, AN = anxiety, HO = hostility, PA = phobic anxiety, PI = paranoid ideation, and PS = psychoticism, GSI = Global Severity Index, COV = COVID. p(1,2) denotes the relative effect of the two independent samples 1 (male) and 2 (female) such that p(1,2) > 1/2 implies that data in group 2 (female) tend to be larger than those in group 1 (male) and vice versa.

**Table 5 ijerph-19-08261-t005:** Correlation coefficients between the COVID-19 impact and psychological distress.

	BSI	Country				
Gender	Subscale	Sweden	Morocco	Serbia	Vietnam	USA
Male	SO	−0.17 *	0.17 *	0.21 *	0.12 *	−0.07
	OC	−0.21 *	0.29 *	0.15 *	0.12 *	0.24
	IS	−0.16 *	0.35 *	0.21 *	0.12 *	0.23
	DE	−0.29 *	0.33 *	0.23 *	0.11 *	0.20
	AN	−0.27 *	0.21 *	0.20 *	0.11 *	0.19
	PA	−0.20 *	0.30 *	0.24 *	0.14 *	0.00
	PI	−0.17 *	0.29 *	0.16 *	0.09 *	0.24
	PS	−0.17 *	0.29 *	0.15 *	0.15 *	0.19
	GSI	−0.25 *	0.31 *	0.20 *	0.13 *	0.19
Female	SO	−0.12 *	0.42 *	0.31 *	0.12 *	0.15 *
	OC	−0.19 *	0.34 *	0.35 *	0.14 *	0.16 *
	IS	−0.13 *	0.33 *	0.31 *	0.11 *	0.04
	DE	−0.26 *	0.37 *	0.39 *	0.12 *	0.24 *
	AN	−0.21 *	0.38 *	0.36 *	0.16 *	0.18 *
	PA	−0.17 *	0.22 *	0.26 *	0.10 *	0.19 *
	PI	−0.15 *	0.37 *	0.32 *	0.10 *	0.19 *
	PS	−0.20 *	0.32 *	0.33 *	0.12 *	0.23 *
	GSI	−0.22 *	0.40 *	0.39 *	0.14 *	0.22 *

Notes. BSI = Brief Symptom Inventory, SO = somatization, OC = obsession–compulsion, IS = interpersonal sensitivity, DE = depression, AN = anxiety, HO = hostility, PA = phobic anxiety, PI = paranoid ideation, and PS = psychoticism, GSI = Global Severity Index. * Significance level = 0.05.

**Table 6 ijerph-19-08261-t006:** Predicting Global Severity Index from the COVID-19 impact score and gender.

		* β *	* se *	* T *	* β **p*-Value	*R* ^2^	* F *	*p*-Value
Sweden	Intercept	1.71	0.04	39.93	<0.001	0.05	26.31	<0.001
	COVID-19 impact	−0.08	0.01	−7.17	<0.001			
	Gender (female)	0.27	0.2	1.32	<0.187			
Morocco	Intercept	0.91	0.10	9.49	<0.001	0.17	31.82	<0.001
	COVID-19 impact	0.12	0.02	7.72	<0.001			
	Gender (female)	0.79	0.38	2.07	0.039			
Serbia	Intercept	0.57	0.06	9.64	<0.001	0.13	54	<0.001
	COVID-19 impact	0.10	0.01	10.32	<0.001			
	Gender (female)	0.26	0.29	0.89	0.373			
Vietnam	Intercept	0.80	0.05	16.1	<0.001	0.03	10.23	<0.001
	COVID-19 impact	0.03	0.01	3.52	<0.001			
	Gender (female)	0.53	0.18	2.86	0.004			
USA	Intercept	1.28	0.18	7.22	<0.001	0.06	7.01	<0.01
	COVID-19 impact	0.07	0.02	3.25	<0.001			
	Gender (female)	0.35	0.20	1.75	0.081			

Notes. *β* denotes the correlation coefficients, *se* denotes the standard error, and *R*^2^ denotes the multiple coefficients of determination.

## Data Availability

Original data tables used for statistical analyses are available by contacting the principal investigator (NK).

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
