# Peer review of "The COVID-19 Pandemic and Adolescents’ Psychological Distress: A Multinational Cross-Sectional Study"

_ijerph, 2022, doi:10.3390/ijerph19148261_

Round 1

Reviewer 1 Report

I think that the article is well written. I have some minor comments (citing the respective line numbers)

L97: There seems to be a faulty title of chapter 1.4. The chapter is titled "Interpersonal sensitivity", which duplicates the previous chapter. The chapter is about depression.

Chapter 2.3: I am missing the information on the estimator used in the invariance analysis.

L262: It would be better to report both alpha and omega as measures of reliability in line with Flora 2020 (https://journals.sagepub.com/doi/full/10.1177/2515245920951747). 

Reviewer 2 Report

Assessment of the work The COVID-19 Pandemic and Adolescents’ Psychological Distress: A Multinational Cross-sectional Study.

First of all, congratulations to the authors of the article for their work. However, the present reviewer has some considerations with respect to it that must be taken into consideration.

Although in the introduction section it is well justified and referenced, I consider that when the authors talk about the objective of the work, at the end of the introduction they should introduce what the hypotheses of their work are. This means rewriting part of the discussion section.

With regard to the method section, I consider that the section describing the sample should be included first and then the Measures section. In addition, a procedure section must be included, although the authors refer to reference number 52 regarding this. In this sense, the following can be said: the article with reference 52 and the current one are different, although they seem to use the same procedure, but not the same sample. This leads me to ask why the sample is different. That is, in the case of USA in an article there is an n and in the current article there is another different n. Why subjects are removed. In addition, there is no record of how the data of minors was obtained, where the authorization of the parents is collected, for example.

They are two different articles, so a procedure section must be included.

Regarding the sample, the size of the sample from the USA is striking. I consider that this sample is very small if we compare it with samples from other countries. In relation to the sample, it is very feminized, if we take into account the percentages of men and women, especially in the USA and Serbia. In relation to this, the authors should have made a greater effort to increase the sample size in the USA, and to increase the number of men in the USA and in Serbia. This is a limitation of the study.

On the other hand, the location of the tables in the article does not help the reader. The tables appear in previous epigraphs and are then commented on in the subsequent epigraph. For example, table 2 should be included in fixture 3.1. This occurs in all subsequent tables.

Finally, the reference number 83 does not appear in the text but it does appear in the references section. This should be corrected.

Reviewer 3 Report

1.Lack of discussion

The authors made a good summary of the analyses results, but there is not enough discussion regarding cross-national differences. Simple citing studies that have various findings is not enough. It will be helpful to expand discussion with recent literature.

2. Sample

Please provide more information regarding sample characteristics.

3. Validity of measures.

Please provide more information regarding variable used.

Are scales used in this study validated for adolescent students in all cross-national samples as well?
